# Sensitive Room-Temperature Graphene Photothermoelectric Terahertz Detector Based on Asymmetric Antenna Coupling Structure

**DOI:** 10.3390/s23063249

**Published:** 2023-03-19

**Authors:** Liang Hong, Lanxia Wang, Miao Cai, Yifan Yao, Xuguang Guo, Yiming Zhu

**Affiliations:** 1Terahertz Technology Innovation Research Institute, University of Shanghai for Science and Technology, Shanghai 200093, China; 2Shanghai Institute of Intelligent Science and Technology, Tongji University, Shanghai 200092, China

**Keywords:** graphene, photothermoelectric, terahertz detector, noise equivalent power

## Abstract

A highly sensitive room-temperature graphene photothermoelectric terahertz detector, with an efficient optical coupling structure of asymmetric logarithmic antenna, was fabricated by planar micro-nano processing technology and two-dimensional material transfer techniques. The designed logarithmic antenna acts as an optical coupling structure to effectively localize the incident terahertz waves at the source end, thus forming a temperature gradient in the device channel and inducing the thermoelectric terahertz response. At zero bias, the device has a high photoresponsivity of 1.54 A/W, a noise equivalent power of 19.8 pW/Hz^1/2^, and a response time of 900 ns at 105 GHz. Through qualitative analysis of the response mechanism of graphene PTE devices, we find that the electrode-induced doping of graphene channel near the metal-graphene contacts play a key role in the terahertz PTE response. This work provides an effective way to realize high sensitivity terahertz detectors at room temperature.

## 1. Introduction

The terahertz (THz) radiation (0.1–10 THz) is located in the transition area between electronics and optics and includes a wide electromagnetic frequency spectrum between microwaves and infrared, and its optoelectronic hybrid characteristics have a wide range of potential applications, including wireless communication, spectroscopy, material science, sensing and imaging, etc. [1,2,3]. Since the 1980s, the rapid development of ultrafast lasers and semiconductor technology has greatly facilitated the development of terahertz technology. Terahertz detectors are key components of terahertz systems, but the low terahertz photon energy makes it an extremely challenging task to achieve high-speed, sensitive terahertz detection. Because the terahertz frequency band is located in the frequency gap between microwaves and infrared, there are unprecedented methods for bridging the gap. Terahertz detectors should have high sensitivity, short response time, low power consumption, compatibility with readout circuits, and low manufacturing cost. Currently, widely used terahertz detectors such as thermoelectric/Golay cell detectors, Schottky diode detectors, quantum well detectors, bolometers, and field-effect transistor detectors, all have some limitations, such as low sensitivity, complex material and device fabrication processes, or low-temperature operation mode [4,5,6,7]. In order to expand the application range of terahertz detectors, it is necessary to develop terahertz detectors that can work at room temperature and have high sensitivity and fast response time.

Photothermoelectric (PTE) detectors are a kind of optical-thermal detector based on the PTE effect. Unlike photoconductive and photovoltaic detectors whose spectral response range is limited by the active semiconductor bandgap used in the detector, PTE detectors are known for their ultra-wideband response operation at room temperature and zero driven bias [8,9,10,11,12]. In addition, the PTE detectors have the advantages of low noise, no cooling unit, no external power supply, DC/AC dual-mode operation, and the ability of detecting infrared and terahertz radiations. PTE can be divided into two processes: photothermal conversion and thermoelectric effect. For photothermal conversion, carriers absorb terahertz photons and transfer the photon energy to the other carriers through electron-electron interaction, which results in the non-equilibrium distribution of carrier gas and the temperature gradient along the device channel. The carrier temperature gradients caused by terahertz photon absorption drive the diffusion of carriers from the hot end to the cold end and create potential differences. Considering the variation of one-dimensional temperature and Seebeck coefficient, the photogenerated voltage (V_PTE_) of PTE detector is the integral of the product of Seebeck coefficient and temperature gradient along the conduction channel of the device. The formula is
(1)$$V_{\mathrm{PTE}}=\int S(x)\;\Delta T(x)$$
where S(x) is Seebeck coefficient and ΔT(x) is temperature gradient. The Seebeck coefficient S reflects the magnitude of the temperature difference electric potential per unit temperature change; when k_B_T << E_F_, according to Mott equation, S can also be written as
(2)$$S=-\frac{\pi^{2}k_{B}{}^{2}T}{3e}\frac{1}{\sigma }\frac{d\sigma }{{\mathrm{dE}}_{F}}$$
where k_B_ is Boltzmann constant, e is the electron charge, σ is the electrical conductivity of the material, E_F_ is Electron energy. From Equation (2), S is proportional to the conductivity and the derivative of energy, so the Seebeck coefficient can be regulated by adjusting the Fermi energy level, which is generally divided into gate control and chemical doping. For PTE detectors, if we assume that the Seebeck coefficient varies in a small range along the channel, the asymmetric distribution of temperature is necessary. 

Since the discovery of graphene in 2004, researchers have begun to study various 2D materials in depth, and the potential use of these materials as photodetectors in a wide range of electromagnetic spectrum is an important research direction [13,14,15,16,17,18,19]. The intrinsic bandgapless graphene has strong absorption at all frequencies, with spectral responses ranging from visible to terahertz bands. The electron heat capacity of few-layer graphene is much lower than that of bulk materials, resulting in the greater temperature variation with the same absorbed energy [20,21,22]. The remarkable thermoelectric effect of two-dimensional materials such as graphene and black phosphorus shows a good prospect in the development of high performance room temperature terahertz detectors [23,24]. For example, 2D material-based PTE terahertz detectors were built by depositing different metallic materials as electrodes [25]. In these devices, electrode-induced channel P-type and N-type doping results in an asymmetric distribution of Seebeck coefficients along the device channel, which induces a PTE response. In addition, the split gate was used to construct lateral PN junction in the device channel to achieve efficient terahertz PTE response [26]. Although excellent device performance is achieved, the above device structures are relatively complex. Additionally, the gate voltage may introduce excess noise. 

In this work, a simple antenna coupled asymmetric structure has been used to achieve the excellent device performance of graphene PTE detectors, which can be compared with the state-of-the-art two-dimensional materials terahertz detectors. Furthermore, we propose a qualitative model to analyze the highly sensitive PTE response caused by asymmetric coupling. In particular, we found that the metal-graphene contacts play a key role in the terahertz PTE response. Our results have important guiding value for further optimization of PTE terahertz detectors for two-dimensional materials in the future.

## 2. Materials and Methods

### 2.1. Device Design and Fabrication 

We used FDTD electromagnetic simulation to simulate and optimize the antenna coupling structure of the detector, which has excellent local field enhancement for far-field terahertz waves. Because high-resistivity silicon has high terahertz transmittance and can effectively reduce the reflection of terahertz waves, a substrate of high-resistivity silicon (ρ ≈ 20,000 Ω cm) covered with 300 nm SiO_2_ was used for the construction of graphene PTE detectors. The detectors were fabricated by standard planar micro-nano device processes and two-dimensional material transfer techniques. Firstly, the coupling structure of the metal antenna (5 nm Ti/40 nm Au) was prefabricated by lithography, electron beam evaporation, and lift-off processes. Then, the highly oriented pyrolytic graphite was mechanically dissociated into micron-sized few-layer graphene (~30 nm) using blue film tape, and the graphene flake was accurately transferred to the channel area of the device by using 3D mechanical transfer platform and PDMS-assisted transfer technology. The graphene device is encapsulated with metal leads onto a custom PCB board for subsequent electrical and optical measurement.

### 2.2. Device Characterizaions

The output characteristic curve of the device was firstly measured using a high-precision digital source meter (Keithley 2400). The SMB100A microwave source (110 kHz–20 GHz) was used to drive a VDI solid-state 6-fold frequency multiplier to produce high-frequency radiation (75 GHz–110 GHz) with a power density of 5 mW/cm^2^. The output port of the multiplier is waveguide-coupled via a standard gain horn antenna, which emits linearly polarized high-frequency radiation into free space. The fundamental frequency microwaves are electrically modulated, and then electrically modulated terahertz radiation is output from the VDI multiplier and irradiated onto the device. In detail, a terahertz beam electrically modulated with a frequency of 1 kHz is irradiated onto the detector, and the generated photocurrent signals are amplified by a low-noise current amplifier (SR570) and then recorded by an oscilloscope (Tektronix MSO2024B). The noise spectrum of the detector is obtained by the spectrum analyzer (Keysight N9322C). The test process of the device was all carried out in room temperature and air environment. 

## 3. Results and Discussion

In order to enhance the PTE response of graphene to improve the performance of terahertz detection, it is necessary to design an efficient asymmetric optical coupling structure. We designed a logarithmic antenna coupling structure which can make the local enhancement region of terahertz field at the source terminal of the device. Figure 1a,b show the overall structure and the channel region of the optical coupling structure of high-performance graphene PTE detector with asymmetric temperature distribution. The logarithmic antenna consists of two metal sleeves containing a plurality of circular tooth strips. The geometric design rules follow the so-called spoof–plasmon effect, which converts the incident electromagnetic field into a surface plasma-induced local field, which causes the terahertz light to focus into the channel region of the device [27]. The coupled antenna is placed on the side close to the source end to form an asymmetric configuration, with two vertically placed metal strips as readout electrodes to collect the photoelectric signals. Since more terahertz electromagnetic energy is concentrated on the source side, unilateral heating along the graphene channel creates an effective temperature difference, enabling the PTE terahertz detection. The field distribution characteristics along the channel direction at different frequencies were obtained by FDTD simulation (Figure 1c). It can be found that the terahertz electric field has strong local enhancement and presents an effective asymmetric distribution. Furthermore, the x-y plane distribution of the electric field intensities in the frequency range of 80~110 GHz further verify the good field enhancement performance of the device (Figure 1d–g), showing the efficient electromagnetic coupling capability of the designed structure.

Figure 2a is the schematic diagram of the terahertz test system. Figure 2b is an optical image of the graphene PTE detector, showing that the prepared device has a complete and good device structure. Good ohmic contact is the key to obtain excellent performance of devices. As shown in Figure 2c, the linear current–voltage (I–V) curve indicates that the device has excellent ohmic contacts. In order to characterize the spectral response of the PTE detector, we measured the photoresponse of the device in the frequency range 75–110 GHz at zero bias voltage. As shown in Figure 2d, the oscilloscope recorded the rapid and stable pulsed photoresponse at different incident terahertz frequencies. It can be seen that the graphene shows excellent optical response performance at different frequencies, and its photocurrent could reach 13.6 µA at 105 GHz. The large photoresponse is due to the excellent thermoelectric properties of graphene at room temperature and the electromagnetic gain of the optimized antenna coupling asymmetric structure. Current responsivity (R_A_) is an important parameter to evaluate the performance of the detector. Its calculation formula is
(3)$$R_{A}=\frac{I_{\mathrm{ph}}}{P_{\mathrm{in}}\cdot A}$$
where I_ph_ is the photocurrent signal of the detector, P_in_ is the power density of the terahertz beam, and A is the area of the detector. In this measurement, the power density of the terahertz beam is 5 mW/cm^2^, and the area of the detector is 600 µm × 300 µm (including the antenna). As shown in Figure 2e, we calculate the zero-bias photoresponsivity of the detector at different frequencies. The results show that the zero-bias photoresponsivity of the graphene device can reach 1.54 A/W at 105 GHz, showing excellent device performance. In addition, we further verify the reliability of the photocurrent signal of the device at zero bias voltage. As shown in Figure 2f, we can find that when the terahertz wave impinges on the device, the output current increases instantaneously. The current in the on state corresponds to the highest point of the photocurrent waveform and the current in the off state corresponds to the lowest point, and the difference between the two current corresponds to the magnitude of the photocurrent.

In the following, we discuss on the speed of the PTE detector, which is usually defined as the time measured from 10% up to 90% on rising edge of signal as well as the recovery time (from 90% down to 10% of the falling edge). As shown in Figure 3a, the response/recovery times of the device were 900 ns and 1.2 µs in a single signal period, respectively. The fast response speed of the graphene device is due to the high mobility of hot electrons in graphene, which is faster than typical room temperature thermal detectors (such as pyroelectric detectors and bolometers), which is advantageous for practical applications. In addition, by setting different electrical modulation frequencies of the terahertz source, we recorded the photoresponse of the graphene device at the terahertz frequency of 100 GHz. As shown in Figure 3b, the photocurrent of the graphene device does not decrease significantly in the modulation frequency range of 0–20 kHz (limited by the maximum electrical modulation frequency of the terahertz source), further confirming the high sensitivity and high-speed response capability. Figure 3c shows the stable waveforms of the device recorded by the oscilloscope at 10 kHz and 20 kHz modulation frequencies. An important characteristic of the ideal terahertz detector is the linear dynamic range, that is, there is a linear relation between input terahertz intensity and the photocurrent. Figure 3d shows a good linear relationship between photocurrent and incident power, confirming the power dependence of PTE response. From the perspective of practical applications, the noise equivalent power (NEP) is another key parameter to evaluate the performance of photodetectors. NEP is defined as the lowest detectable power within the measurement bandwidth of 1 Hz, calculated by
(4)$$\mathrm{NEP}=\frac{i_{n}}{R_{A}}$$
where i_n_ is the noise value at the modulation frequency, and R_A_ is the responsivity of the detector. The noise spectrum of detectors usually contains several different main noise sources, including 1/f noise, shot noise and thermal noise (Johnson-Nyquist noise) [28,29,30]. The 1/f noise results from changes in electronic state and is prevalent at low frequencies (below 1 kHz). Shot noise is caused by photoexcited carriers randomly generated by the detector under radiation or thermal excitation. Thermal noise is related to the ohmic resistance and temperature of the detector and is generated by the random thermal motion of charge carriers. Since our graphene devices are operated in zero-bias mode, the shot noise is negligible, and the thermal noise dominates the noise spectrum of the detector. In order to obtain the detector sensitivity, we used a spectrum analyzer (Keysight N9322C) to obtain the noise spectrum of the devices under zero bias (Figure 3e). As shown in Figure 3f, we evaluated the NEP of the graphene device at different frequencies. The terahertz graphene PTE detector achieved a low NEP of 19.8 pW/Hz^1/2^ at 105 GHz, showing good detection sensitivity. The performance comparison of the 2D material-based terahertz detectors is shown in Table 1, showing that our graphene PTE devices have good performance. 

In order to optimize the device performance in the future, we proposed a qualitative theoretical model to analyze the high terahertz sensitivity caused by the asymmetric antenna coupling structure. As shown in Figure 4a, the IV curves of the graphene device remain parallel under 105 GHz illumination and dark conditions, indicating that the bolometric effect is negligible. In addition, the photoresponse remains constant with the increase in applied bias, which is consistent with the bias insensitivity of the PTE effect [26]. Figure 4b shows the schematic diagram of the response mechanism of the graphene PTE detector. In our device, the coupling antenna and the readout electrode are separated, and the coupling antenna is positioned away from the center of the channel and acts as a focusing structure to heat one end of the graphene. Since the readout electrodes at both ends of the graphene channel are of the same metal, the doping effect of the metal electrode on both ends of the graphene is the same, so it can be considered that the Seebeck coefficients of the doping regions at both ends are the same. Considering the high thermal conductivity of the metal and the fact that the focusing structure of the antenna does not directly heat the electrode, it can be considered that the temperature difference of the source-drain metal readout electrode is negligible. According to Equation (1), the thermoelectric response of the graphene channel can be written as
(5)$$V_{\mathrm{PTE}}{=S}_{1}{(T}_{1}{-T}_{0}{)+S}_{0}(T_{2}{-T}_{1}){+S}_{1}{(T}_{0}{-T}_{2}{)=(S}_{1}{-S}_{0}{)(T}_{1}{-T}_{2})$$
where S_1_ is the Seebeck coefficient in the metal-doped area, S_0_ is the Seebeck coefficient of graphene away from the metal electrode, T_0_ is the metal temperature, T_1_ and T_2_ are the temperature at the end of the doped graphene, and T_h_ is the highest temperature point of channeled graphene. According to Equation (5), the electrode-induced doping of graphene channel near the metal-graphene contacts play a key role in the terahertz PTE response. The thermoelectric response of graphene is determined by the electrode-induced doping on graphene and the temperature difference along graphene channel. Due to the good doping of metals to graphene, this ensures the difference of Seebeck coefficients between the doped and undoped regions of graphene. Considering the contribution of Seebeck coefficient difference, the PTE response will be further improved if the signs of the Seebeck coefficient are opposite. Furthermore, the location of the optical coupling antenna in the graphene channel determines the temperature difference required to achieve the graphene PTE response. It can also be seen that when the coupling antenna is located in the middle of the device channel, due to the symmetrical configuration, the temperature difference will be zero, and the PTE response is completely suppressed. As the coupled antenna moves away from the center of the graphene channel, the temperature at the end of the graphene doping region near the coupled antenna will be closer to the maximum temperature point, and the effective temperature difference to drive the PTE response of the graphene will be expanded, thus improving the graphene PTE response. 

## 4. Conclusions

In summary, we achieved sensitive graphene-based PTE terahertz detectors by using an efficient asymmetric antenna coupling structure, which exhibits excellent zero bias detection performance in the frequency range of 75 GHz to 110 GHz. The measurement results show that the graphene device has a zero bias photoresponsivity of 1.54 A/W, a noise equivalent power of 19.8 pW/Hz^1/2^, and a response time of 900 ns. Furthermore, through qualitative analysis of the response mechanism of graphene PTE devices, we find that the electrode-induced doping of graphene channel near the metal-graphene contacts play a key role in the terahertz PTE response. These results show that the optical coupled structure we designed effectively improves the thermalelectric terahertz detection sensitivity, providing an effective way to further improve the performance of two-dimensional materials terahertz detectors in the future.

## Figures and Tables

**Figure 1 sensors-23-03249-f001:**
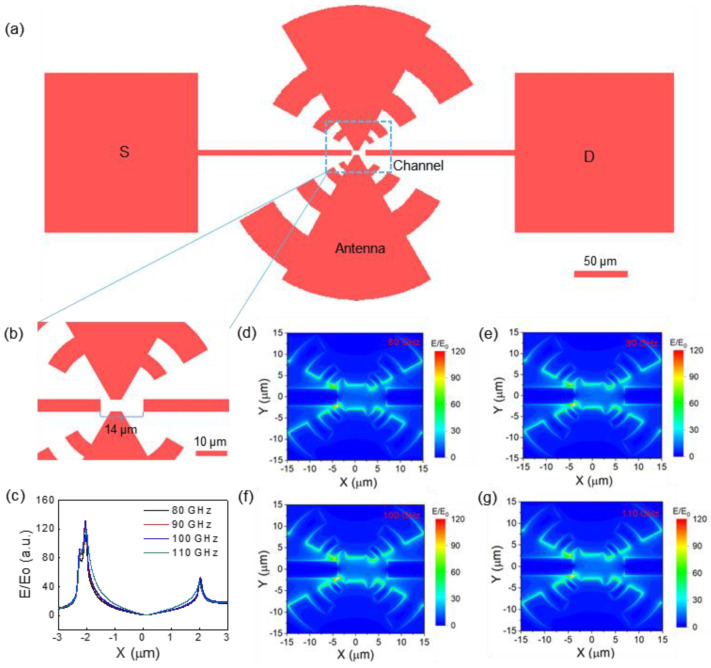
(**a**) Structure of the logarithmic antenna optically coupled PTE device; (**b**) enlarged view of device channel active region; (**c**) the electric field intensity distribution along the channel direction of the coupled structure; (**d**–**g**) electric field distributions at different frequencies in the x–y plane.

**Figure 2 sensors-23-03249-f002:**
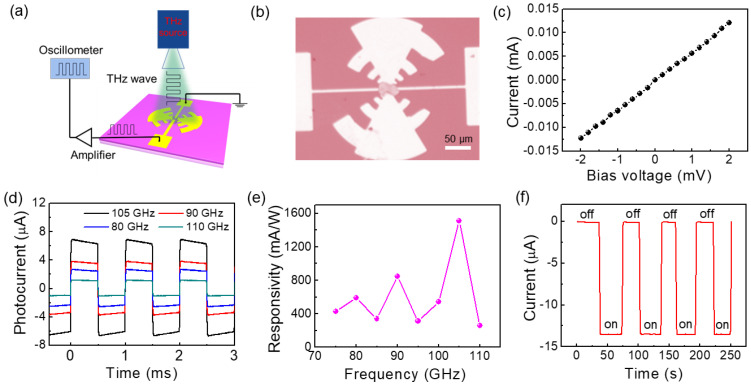
(**a**) Schematic diagram of terahertz photocurrent test system; (**b**) optical image of terahertz graphene PTE detector; (**c**) room-temperature I–V curve; (**d**) photoresponse waveform of the device under different frequency illumination; (**e**) the zero-bias photoresponse of devices at different frequencies; (**f**) real–time output current when the device is fixed at zero–bias voltage at 105 GHz.

**Figure 3 sensors-23-03249-f003:**
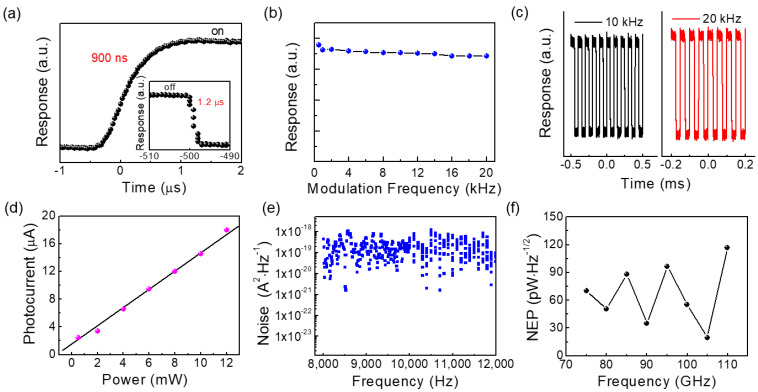
(**a**) Response time of the graphene detector; (**b**) variation of optical response with modulation frequency; (**c**) the optical response waveform of the graphene detector at 10 kHz and 20 kHz modulation frequencies; (**d**) power dependent optical response; (**e**) measured noise spectral density of the graphene detector at 300 K; (**f**) noise equivalent power of the graphene detector at different frequencies.

**Figure 4 sensors-23-03249-f004:**
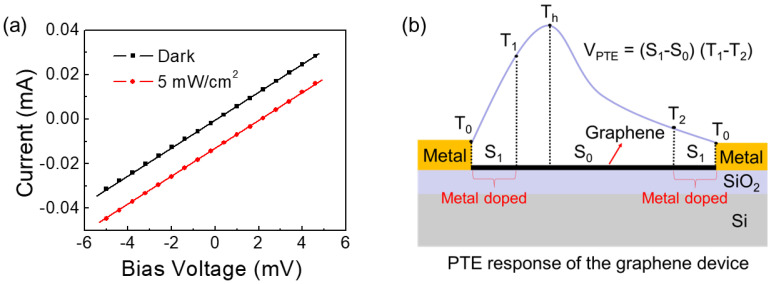
(**a**) The I–V curve of the detector with and without terahertz illumination; (**b**) Schematic diagram of the photothermoelectric response of graphene.

**Table 1 sensors-23-03249-t001:** Comparison of performance of 2D material-based terahertz detectors.

Material	Frequency	Responsivity	NEP	Response Time	Reference
black phosphorus	0.29 THz	135 V/W	138 pW/Hz^1/2^	800 ns	[24]
Graphene	3 THz	49 V/W	160 pW/Hz^1/2^	3 ns	[31]
Graphene	0.6 THz	764 V/W	515 pW/Hz^1/2^	-	[32]
PdSe_2_	0.1 THz	0.02 A/W	142 pW/Hz^1/2^	7.5 μs	[33]
Graphene	0.105 THz	1.54 A/W	19.8 pW/Hz^1/2^	900 ns	This work

## Data Availability

Not applicable.

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
