# Peer review of "Sensitive Room-Temperature Graphene Photothermoelectric Terahertz Detector Based on Asymmetric Antenna Coupling Structure"

_sensors, 2023, doi:10.3390/s23063249_

Round 1

Reviewer 1 Report

Authors are recommended to revise the paper as per the following comments:

1. The novelty of the work is not clear. Authors should clearly mention the motivation behind this work and the novelty of the proposed work.

2. Authors should improve the state of the art analysis by discussing more recently reported works (Specifically from the year 2021 and 2022).

3. Authors should include a comparison with related works reported recently (especially from years 2021 and 2022), preferably in the form of a table to emphasize the contribution of this proposed research work.

4. Authors are recommended to add the measurement setup details in the manuscript.

5. Authors should check and correct the grammatical mistakes. 

Reviewer 2 Report

This paper reports the fabrication of graphene based photothermal THz

detector and its obtained sensing performance. This subject would be

interesting for THz people, but you are requested to respond to the

following points in the text:

1.    Compare your obtained sensing performance regarding NEP, response time, optical response band, spatial resolution, fabrication feasibility etc. in concrete quantitative comparison with known other THz detectors, e.g. by means of table or something like.

2.    What do you mean by “short” noise? It looks like “shot” noise. If so, since the shot noise depends on receiving light power, the entire noise has to be more properly quantitatively evaluated.

3.    As a whole, this paper is well represented, but your original contributing points that you want to claim for are not well distinguished. Please clarify them in the text.

Reviewer 3 Report

About the experiment:

1.       What is the beam profile of the THz source? Is that a horn antenna on the source in Figure1(a)?

2.       Is the detector sensitive to incident THz polarization?  What is the polarization of the source?  

3.       You need to the list at least three other works about graphene photothermoelectric THz detector in a table to compare all performances to show the advantage of your design.

Simulation:

1.       The simulation section could support your experiment better. For example: how do you decide the distance of gap of channel? Or how much does the gap deviate from the center?

2.       You need a scale bar on Figure 1 (a) and (b); 

3.       The Figure 1. (d)-(g) are very dark and you can make them clearer and more informative by tuning the heatmap color contrast. 1) you should keep them the same scale and shape of the figure1(b). 2) add the units on the color bar. 3) the labeled frequencies are hard to read

Reviewer 4 Report

The paper reports a novel type of photothermoelectric Terahertz Detector. The paper is technically sound, meticulous, and clearly presented.

Just one minor amendment is required. Figs 2c, 2e, 3b, 3e, 3f, 4b require error bars.

Round 2

Reviewer 2 Report

The authors' responses to my previous questions and comments are satisfactory to me.